# Bring It to an End: Does Telomeres Size Matter?

**DOI:** 10.3390/cells8010030

**Published:** 2019-01-08

**Authors:** Camille Laberthonnière, Frédérique Magdinier, Jérôme D. Robin

**Affiliations:** Aix Marseille Univ, MMG, Marseille Medical Genetics U1251, 13385 Marseille, France; camille.laberthonniere@univ-amu.fr (C.L.); frederique.magdinier@univ-amu.fr (F.M.)

**Keywords:** telomere, TPE, TPE-OLD, TERRA, aging

## Abstract

Telomeres are unique nucleoprotein structures. Found at the edge of each chromosome, their main purpose is to mask DNA ends from the DNA-repair machinery by formation of protective loops. Through life and cell divisions, telomeres shorten and bring cells closer to either cell proliferation crisis or senescence. Beyond this mitotic clock role attributed to the need for telomere to be maintained over a critical length, the very tip of our DNA has been shown to impact transcription by position effect. TPE and a long-reach counterpart, TPE-OLD, are mechanisms recently described in human biology. Still in infancy, the mechanism of action of these processes and their respective genome wide impact remain to be resolved. In this review, we will discuss recent findings on telomere dynamics, TPE, TPE-OLD, and lessons learnt from model organisms.

## 1. Introduction

Unlike circular DNA found in prokaryotes, eukaryotes share minisatellite sequences at the end of linear chromosomes named telomeres. The importance of this particular structure was first demonstrated in the late 1930s by McClintock and Muller [1]. By irradiating *Drosophila melanogaster* fruit flies, they observed that mutations never occurred at the end of the DNA molecule suggesting the presence of a protective cap restricted to these regions. Since then, explorations and hypothesis on their complete function never ceased within the scientific community and beyond (i.e., general audience).

Major findings on the organization of telomeres were obtained in the *Tetrahymena* ciliate model, which possess hundreds of mini telomeres, with the identification of its nucleotide composition [2] and the deciphering of the telomerase-dependent mechanism involved in extending this hexameric sequence [3,4,5].

Thanks to the increased sensibility of techniques, the human telomeric sequence composed of a 5′-TTAGGG-3′ repeated motif was later identified [6] followed by the identification of the particular 3′ overhang and T-loops structure [7,8,9] bound by proteins forming the sheltering complex [10,11,12,13,14,15,16,17,18,19]. Altogether, the nucleoprotein structure creates the protective cap hypothesized in the 1940s, and set the molecular base of subsequent telomere studies.

The genetic information of eukaryotes is assembled in a linear genome. Hence, the first major role of DNA ends is to protect informative genetic sequences from erosion. Free DNA extremities are exposed to exonuclease and at risk for non-homologous ends joining between DNA molecules [20]. These regions are thus considered as endangered sequences for the cell. Telomere structure limits this loss and preserves the integrity of chromosomes while allowing distinction between natural chromosome ends and damaged DNA that need to be repaired [21,22]. 

From cancer research to aging studies, investigations on the regulation of the very tip of chromosomes led to paradoxical observations: (i) telomerase, the enzyme responsible for telomere elongation is not produced in somatic cells, (ii) most cancer telomerase-positive cells display rather short telomeres as telomerase upregulation is considered a late event during malignant transformation; (iii) stem cells harbors long telomeres and express telomerase; (iv) short telomeres are associated with aging and increased cancer risk, or the complete opposite; (v) telomere maintenance involves epigenetic mechanisms and chromatin changes. Altogether, the wealth of studies available in the literature raises questions on the relevance, feasibility, and caveats that one encounters when questioning telomeres dynamics. In the end, does telomere length and size matter?

In this review, we will discuss the past decades of telomere research by crossing lessons on telomere length regulation and impact in Human. We will first focus on telomere length dynamics before developing on consequences at the epigenetic level and possible implication on human health through position effect mechanisms. 

## 2. Telomere Dynamics

### 2.1. Telomere Shortening

As previously mentioned, DNA extremities are lost through cell divisions. This mechanical DNA erosion is the consequence of incomplete duplication of the lagging strand (e.g., 5′ end of the native strand) during synthesis. Indeed, during replication, the DNA polymerase fails to generate a completed 3′ end in the absence of RNA primer [23,24]. Telomeres shorten in human somatic cells lacking telomerase at a rate of 50–100 bps per population doubling thus predicting the remaining cellular life span of the cell [25,26]. Thanks to this end-replication problem, telomere acts as a mitotic clock, counting each cell division.

Single stranded DNA or free double stranded extremities are toxic for the cell. The shelterin complex protects telomeric DNA to prevent recognition as damaged DNA [27]. Defect in formation of the telomere protective cap leads to cell cycle arrest and is often associated with end-to-end chromosome fusion by non-homologous ends joining (reviewed in [21]). When telomere length reaches a critical limit, chromosome ends become unprotected and are recognized as a double-stranded break DSB [28]. Recruitment of 53BP1, MRE11, phosphorylated histone H2A.X (γH2A.X) and ATM [29] can be visualized in vivo by presence of DSB repair complexes co-localizing with telomeres, dubbed telomere dysfunction-induced foci (TIFs). Recruitment of DDR factors activates a signaling cascade, through ATM-dependent and -independent pathways leading to cell-cycle arrest and replicative senescence (Figure 1). Cells can remain in this state for an extensive period of time in absence of other changes, representing a tumor suppressor pathway for long-lived organisms like humans. Telomere-induced senescence depends on the rate of telomere shortening, initial telomere length and length of the shortest telomeres [30] and can be skipped upon dysfunction of the cellular checkpoints such as p53 or RB. Moreover, studies suggest that not one but a subset of critically short telomeres are required to engage cells toward senescence [31,32]. Using a ChIP-on chip approach targeting γH2A.X histones, a first study deciphered the arm-specific sensitivity of chromosomes to DNA damage response with a higher frequency of damages in a subset of chromosomes ends [32]. Later, another group confirmed this heterogeneity and found similar results in a different model (e.g., 6p, 9q, 15q, 17q, 21p) [31], advocating for a targeted telomere shortening process. Nevertheless, to settle between regulated or stochastic telomere erosion, further studies are required, involving a higher depth of analysis (i.e., in cell types, single telomere ends analysis confirming IF assays, spectrum of epigenetic marks).

For a long time, telomere erosion has been associated with the proliferative state of tissues (e.g., proliferative or static). However, a recent cross study exploiting samples from 87 well-characterized individuals did not sustain this dogma [33]. Indeed, telomere erosion occurs in proliferative tissues such as the skin (and blood), but also in organs considered as post-mitotic with a minimum regeneration rate such as skeletal muscle, heart, or adipose tissue. Even taking into account high regeneration processes happening after muscle injuries or diseases for instance, similar erosion rates were observed. To date, telomere shortening in post mitotic tissue has been poorly investigated and a variety of factors might be implicated such as an increase in reactive oxygen species (ROS), exhaustion of the pool of stem cells required for tissue regeneration or implication of components of the shelterin complex. This also suggests similar telomere lengths, independently of the tissue’s state, an observation reminiscent to the previously proposed average homogeneity of telomere length [34].

Telomere attrition is involved both in normal and pathological aging whereas re-activation of telomerase is able to revert aging in mice [35] or in vitro in cultured cells [36], without provoking malignant transformation. Hence, if critically short telomeres trigger senescence and crisis, critically long telomeres have not yet been described. Nevertheless, a recent study identified TZAP, a telomere-bound protein that acts to trim long telomeres, thus advocating for a maximal regulated length that can be reached under control [37] (Figure 1).

### 2.2. Telomere Elongation

Where telomere shortening can be seen as a tumor-suppressor pathway through replicative senescence (e.g., preventing accumulation of mutations through unlimited divisions), the line between tumor-suppressing and tumor-promoting is thin. At senescence, cell proliferation is inhibited. In the presence of cancer-initiating changes, senescence can be bypassed and cell lifespan extended, a stage called crisis. During crisis, signals to undergo replicative senescence or continue cell divisions are in balance with chromosome end-to-end fusions, accumulation of chromosome abnormalities and massive apoptosis [39]. To acquire cell immortality and survive to crisis, cells with critically short telomeres need to preserve or elongate their telomeres either by reactivating telomerase or by a telomerase-independent mechanism dubbed ALT for Alternative Lengthening of Telomeres [40] (Figure 1).

Telomerase is the specific enzyme that allows such elongation, both by providing the template and catalytic unit. It is composed of the hTERT (human telomerase reverse transcriptase) catalytic subunit and the hTR (human telomerase RNA), a RNA component serving as a primer and template for DNA synthesis. The *hTERT* gene is 42 kb long and located in the subtelomeric region of the short arm of chromosome 5 (5p13.3). TERT promoter is unconventional with a complex regulation dynamics still under investigations [41,42]. Reactivation of telomerase expression occurs in the vast majority of cancer (85%) [43]. In the majority of cases, reactivation is associated with *TERT* promoter mutations, including recurrent mutations identified in 19% of cancers [44].

Telomeres can also be maintained in cancer cells through the telomerase-independent ALT pathway. Human ALT tumor cells exhibit specific characteristics including heterogeneous telomere length, presence of extrachromosomal telomeric repeats (ECTR), C-circles, ALT-associated promyelocytic leukemia (PML) bodies (APBs) and elevated frequency of telomere sister chromatid exchanges (TSCEs) [45]. ALT is a homologous recombination (HR) based telomere maintenance mechanism utilized in around 5 to 15% of human cancers. Understanding of ALT mechanisms is still incomplete but increased DNA damage, due to replication fork collapse or spontaneous telomere break, promotes intra- and/or inter-telomere recombination and telomere extension [46].

As in cancer cells, telomeres need to be finely regulated in stem cells, a cell type where telomerase is constitutively expressed albeit to a lower extent. In pluripotent stem cells (PSCs), in addition to being essential for the maintenance of telomere length, telomerase is required for long-term self-renewal and pluripotency [47]. Use of ALT pathway has been showed in mouse embryonic stem cells (mESCs), especially through activation of the *Zscan4* gene. Its activation is associated with rapid telomere extension by telomere recombination [48]. Human embryonic stem cells (hESCs) and human induced pluripotent stem cells (hIPSCs) do not rely on the ALT pathway for telomere maintenance but only on a high telomerase activity [49]. These differences between human and murine pluripotent stem cells might be due to the naïve state of mESCs and the presumably equivalent to primed state of hIPSc and hESCs. In addition, this variance may reflect the global differences in the biology of telomeres between human and mice.

Adult stem cells are long-lived and actively cycling stem cells. The importance of telomerase in adult stem cells compartment is directly observable in patients suffering from a number of syndromes associated with dysfunctional telomeres that will be discussed below.

### 2.3. Heritable Telomere-Length

It is now widely admitted that telomere length is variable from one individual to another and even variable between each chromosome and their respective arms within one cell. This observation raises questions on telomere length heritability for the 92 telomere ends. Are they partially or totally inherited? Does the age of the parents (i.e., telomere length in gametes) matter? Similarly, does the parental overall telomere length matter? If answers can be grasped from a few studies including some focusing on twins, work on these fascinating questions remains to be done.

Using Fluorescence In-Situ Hybridization (FISH), a group had previously shown some differences in telomeres length between homologous chromosomes [50], suggesting homolog-specific telomere length polymorphisms. In monozygotic twins, the same team elegantly showed highly similar arm-specific chromosome ends between twins and close telomere lengths at advanced age, a similarity not found in dizygotic twins [51,52]. Telomere length was more conserved between the specific chromosome ends of the twins (if one considers separately chromosome pairs) than for the individual itself (if one considers the chromosome pair). Moreover, samples from 80-year-old twins suggested that telomere length was roughly conserved regardless of environmental stresses, at the opposite of what has been described for epigenetics marks and the epigenetic drift [53]. This suggests that chromosome-specific telomere length is inherited from the parents and confirms the hypothesis of a partially genetically inherited telomere length proposed a long time ago [54]. Furthermore, this study advocates for a limited impact of environmental factors over telomere erosion (if one consider that twins likely lived in different environments over time). Respective chromosomal telomeres within cells are highly heterogeneous. Nevertheless, length distribution between chromosomes arms is conserved between cell types in a given individual [33,55,56].

More recently, parental contribution to inherited telomere length has been proposed. A meta-analysis on nearly 20,000 participants revealed strong correlation between paternal age at conception (PAC) and offspring telomere length. Sperm telomere length increases with age in humans. As a result older fathers’ offspring inherit longer telomeres [57]. In a more recent study, using 3 of the largest available datasets, a group refined the calculations, and incorporated the paternal birth year as a corrective parameter allowing comparison of different datasets since paternal birth year (PBY) was collinear with age and father’s age at birth of the offspring in the three datasets analyzed (e.g., considering a generational effect) [38]. Beyond reconciling data from cross-sectional and longitudinal studies, their outcome revealed that global telomere length shortens in the global population [38].

These different studies show heterogeneity in the length of telomeres between individuals, either because of genetic factors inherited from the parents, age of the father at conception or because of environmental factors specific to each person/generation with a theoretically minimal impact for the later (as calculated in [58]). In this context, ‘critically’ short telomere will not affect individuals at the same age or the same manner, even at the scale of individual chromosomes, each having different lengths. Hence, the harmful effect of telomere shortening and impact of telomere erosion can appear as a societal burden.

## 3. Telomere Singularity: Possible Causes of Differences between Telomere Length

As mentioned previously, critical telomere length and sensitivity to DNA damage signaling has been restrained to a small subset of chromosome ends [31,32]. Those findings made by two separate teams, using replicative senescence as cellular model, brought up new concerns toward telomeres dynamics. If some chromosome ends are more prone to shortening (i.e., 6p, 12p, 17q, 16q, 21p, 3p-q, 9p-q), some are protected from DNA damage signaling (e.g., 21p) [32]. Moreover, chromosomes exhibit differential rates of shortening between long and short arms. Altogether, those preliminary studies brought up the singular identity of each telomere. If not equal in length, are they equal in anything (i.e., dynamics or localization)?

### 3.1. Telomere Nuclear Localization

For a long time, chromosome organization within the cell was thought to be random. In mammalian cells, the exact opposite is now well established [59,60]. Chromosomes are organized in gene-poor and gene-rich regions and position within the nucleus often correlates with gene density, gene-poor regions being more peripheral [59,61]. At numerous loci, localization depends on interactions between genomic loci and the nuclear lamina through formation of lamina-associated domains (LAD) [59]. If found widespread in the nucleus with a preference for the nuclear interior, partners and actors involved in the topological organization of telomeres remain elusive [62]. This distribution is also dynamic and changes dependent on the cellular state (proliferative versus quiescent), a finding reminiscent of recent discoveries on proteins associated to the nuclear periphery and telomeres, either in *cis* (A-type Lamins, LAP2a, BAF1) [63,64,65] and/or in *trans*, through TRF2 interactions (A-type Lamins, LAP2a) [66,67]. Since telomeres are not only found at the nuclear periphery, these regulatory elements (anchoring telomeres to the nuclear envelope) do not alleviate telomere spatial positioning. Telomere-specific localization could depend on subtelomeric regions, which are unique to each chromosome [68]. Indeed, in addition to LADs, specific sequences such as the D4Z4 subtelomeric macrosatellite tether telomeres at the nuclear membrane. Interestingly, if studies regarding the relative position of telomeres to the periphery are available, their counterparts do not exist, as if internal telomere localization was considered as the default position. Of note, as mentioned previously, telomere erosion occurs for all telomeres, but with differences. As shown by single telomere analysis (STELA), erosion rates showed heterogeneity when comparing chromosome ends, with a slower shortening rate but overall shorter length for the 17p [69]. However, regarding single ends, telomere length was homogenous, further advocating for a per chromosome-end tight regulation of erosion, rather than an overall random shortening. Further studies combining STELA single telomere length analysis and telomere position within the nuclei could shed light on the mechanisms explaining why some telomeres are more critical when short than others (e.g., 6p and 21p). The key might reside on the *cis* and *trans* interactions at play in their respective territories.

### 3.2. Telomere Replication

Chromosome folding and subnuclear localization are also influenced in response to gene replication [60]. Considering this, chromosomes do not replicate at the same time during S-phase but have a specific window of replication, with some being replicated early and others during late S-phase without synchronicity between chromosome arms (i.e., p and q arms replicating regarding their nuclear localization rather than genomic identity) [70]. This replication timing is conserved between homologs and individuals. Moreover, like for the rest of chromosomes, there is a tight relationship between the peripheral localization of a telomere and its late replication with correlations between replication timing of telomere, presence of satellite-like repeats or position of telomeres at the nuclear membrane as seen for the short arm of acrocentric chromosomes [68]. It’s tempting to assume that telomere length could influence telomere replication timing, however these two mechanisms seem independent. [70].

In a human model of immortalized cells, (i.e., with reactivation of telomerase) there was no change in the global replication profile of telomeres throughout the immortalization process [71]. This corroborates the observation that translocation of a telomere and its associated subtelomere is sufficient to maintain replication timing, independently of the size of the translocation or the presence of a complex rearrangement. In agreement with findings on telomere position, it appears that subtelomeric regions are needed to control telomere replication timing as observed using fragmented telomere [68,70], independently from centromeric or other chromosomal sequences. Thus, the only parameter associated with changes in telomere replication timing seems to be its radial position within the nucleus. Nevertheless, molecular mechanisms and sequences involved in this process remain unknown. Recent studies have underlined the role of TRF2 in the global replication [72,73] which might bring back replication timing of specific telomeres at the center of interest. Of note, the RIF1 protein (Rap1-interacting-factor) able to bind to Lamin B1 involved in nuclear architecture organization has been implicated in the spatio-temporal regulation of the human genome [74]. However, to date, no relation has been made between RIF1 activity and telomeric replication timing.

Altogether, whereas only a few studies investigated telomeres nuclear localization, the wealth of reports on telomere replication mainly focuses on the molecular mechanisms ruling the specific duplication of the (T_2_AG_3_)_n_ motif, mostly in link with the possibility of development of anti-oncogenic therapies targeting telomerase. Thus, an integrative vision of telomeres within its 3D environment remains to be established to further understand the consequences of nuclear localization on higher-order telomere organization and consequences on transcription or DNA repair of chromosome ends (Figure 2).

## 4. Telomere Epigenetics and Position Effect

Beyond the mitotic clock, the very tip of our DNA has been shown to be epigenetically active. Either by impacting transcription by position effect or by direct transcription. TPE (Telomere Position Effect), a long-reach counterpart, TPE-OLD (Telomere Position Effect Over Long Distances) and TERRA (Telomeric Repeat-containing RNA) have been recently described in Human [75,76,77]. Still in infancy, their mechanism of action and genome wide impacts remain to be resolved.

### 4.1. Epigenetic Signature

Mammalian chromosome ends are enriched in constitutive heterochromatin marks such as di- and trimethylation of H3K9, trimethylation of H4K20 and recruit heterochromatin protein 1 (HP1) isoforms in a telomere length-dependent manner [78]. Thanks to the rise of deep-sequencing, a recent study focused on deciphering the global chromatin signature of telomeres. Considering the existence of internal telomeric sequences (ITS) and other possible telomere-like repeats, reads were stringently filtered-out to keep only ‘pure’ and terminal (T_2_AG_3_)_n_ motifs [79,80]. The analysis of 10 epigenetic marks in >10 cell lines from 10 different groups revealed striking results. Rather than the increase in H3K9me3 previously described [81], telomeres appear poorly enriched in this heterochromatin mark compared to other classical heterochromatic regions (e.g., Sat II, Sat III). H3K9me3 enrichment was only found in the U2OS ALT cell line. Telomeres are rather enriched in H4K20me1 and more surprisingly in H3K27Ac, a histone mark highly linked to active transcription [82]. Reciprocally, telomeres are also depleted in methylated H3K27. These unprecedented findings radically change the dogma for telomeres influence, usually seen as massive silencer and suggests the need for re-exploration of ChIP-Seq data. Indeed, using these refined settings, one can assume new epigenetic marks and proteins associated with telomeres. Additionally, telomere length should be taken into consideration, in order to decipher the potential influence of telomere fluctuations over the genome.

### 4.2. Telomere Position Effect

Telomere length also alters expression of genes located in subtelomeric regions through TPE. This mechanism first revealed in Drosophila was extensively described in *Saccharomyces cerevisiae* [83]. By placing a gene directly downstream of telomeric repeats, it was shown in this model organism that, gene expression could be reversibly repressed in a telomere length-dependent manner. Classical TPE requires the spreading of telomeric heterochromatin to nearby genes and extends to a few kilobases from the telomeres [84] (Figure 3).

TPE resembles Position Effect Variegation (PEV), first described in 1930 in *Drosophila melanogaster* [85]. This silencing mechanism refers to the difference in expression observed when an identical gene is positioned at different sites in the genome. The process relies on chromatin-associated factors, which are able to silence gene expression by spreading to nearby sequences and was characterized through transgenesis. Integration into euchromatin resulted in strong gene expression while integration into heterochromatin resulted in transcriptional repression [86].

The capacity of heterochromatin marks to spread along chromosomes suggested the existence of similar mechanisms at the level of telomeric heterochromatin and lead to the identification of TPE. The first in vivo evidence came from a replication timing assay on chromosome 22 with a heterozygous microdeletion of 130 kb at the end of the 22q arm [87]. The microdeletion resulted in a difference in the replication timing of the two 22q arms. However, both alleles of a gene located at almost 50 kb from the breakpoint were expressed, suggesting a positional effect from telomeres, delaying the replication of the adjacent chromosomal regions. Yet, the existence of telomere effect in human cells was not evident for all. Further evidence supporting this mechanism came from in vitro experiments, using transgenes inserted in the vicinity of telomeres, thanks to telomere-seeding constructs. Insertion of the luciferase reporter adjacent to a newly formed telomere resulted in a 10-fold decreased expression of luciferase compared to controls [75]. Expression was further restored after treatment with a histone deacetylase inhibitor (Trichostatin A), suggesting the involvement of repressive chromatin marks in the process and histone deacetylation, as found in other systems (e.g., flies and yeast). In agreement, stable transfection of a plasmid containing an eGFP reporter followed or not by a 1.6 kb stretch of telomeric repeats, showed that the percentage of EGFP-positive cells was more pronounced in the absence of telomeric repeats [88]. Interestingly, the transient transfection of the same construct, showed that the percentage of EGFP-positive cells was more pronounced in the presence of telomeric repeats. These experiments revealed that TPE does not simply results from the binding of transcriptional repressor to chromatin but regulated by telomeric repeats. The same type of experiments enabled determining the distance over which a telomere can exert its influence. Theoretically, the level of gene repression decreases as the distance of the gene from the telomere increases. Telomeric chromatin marks can spread and repress gene expression to up to 100 kb from the telomere [89,90] with a more pronounced effect when telomere are long. Telomeric silencing also requires methylation of CpGs islands at subtelomeric regions and the recruitment of SIRT6 [91].

Not all subtelomeric genes are susceptible to TPE suggesting the existence of subtelomeric sequences able to counteract TPE by acting as boundaries. Insulators and boundaries are DNA element able to alter gene expression by preventing activation or inhibitory effects due to the chromatin environment [92]. The first discovered insulator protecting from TPE in higher eukaryotes was cHS4, the chicken beta-globin insulator [93,94]. This CTCF-dependent insulator counteracts heterochromatin expansion, leaving the transgene in a more open conformation, allowing deposition of histone methylation marks like H3K4me2 and H3K79me2. Further studies are required to see its transposition in a human system, as a very few sets of TPE genes are known.

Indeed, in human, evidence for the impact of telomere length was verified in young and old cells (with respectively short and long telomeres) by transcriptomic analysis [95]. This work led to the identification of *ISG15* (Interferon Stimulated Gene 15 kDa) as the first TPE gene up-regulated upon telomere shortening. These findings were consistent with yeast TPE mechanisms. In this specie, most subtelomeric genes are involved in metabolism and stress response and are silenced in optimal growth conditions [96]. These genes, such as PAU involved in cell wall and drug resistance [97] or FLO involved in cellular adherence are activated upon stress [98], nutrient starvation or chemical treatment. Interestingly, as observed in budding yeast, the human *ISG15* gene product is involved in the innate immunity response pathway suggesting a conservative role of TPE, as a stress-response and adaptation mechanism [99].

Interestingly, 15 genes are present between this gene and the telomere of the short arm of chromosome 1. None of these genes showed telomere length-dependent regulation, a contradiction with the TPE rule, advocating for higher complexity in the regulation of this mechanism.

### 4.3. Telomere Position Effect-Over Long Distance

If discontinuous TPE has been shown in yeast [100], its existence in mammals is fairly new. One hypothesis explaining TPE discontinuity in mammals might reside in DNA looping as involved in enhancer-blocking activity or *trans*-acting changes occurring elsewhere in response to telomere shortening (Figure 3).

The major difficulty when looking at telomere length-dependent mechanism is the capacity to impute gene expression changes to telomeres only. To overcome this limitation, reversibly immortalized cell lines expressing a *hTERT* (telomerase reverse transcriptase) transgene flanked by LoxP sites were made [90]. After cell cloning, *TERT* can be excised at different time points to generate isogenic clones in which the only significant difference is the length of telomeres after TERT removal (e.g., same time in culture).

This system was applied first to primary myoblasts and fibroblasts. Comparing myoblasts with long and short telomeres, 144 telomere length-sensitive genes were identified, amongst the 1423 subtelomeric genes analyzed. These genes were located as far as 10 Mb from the telomere. Analysis of long-distance interactions revealed the existence of loops between TPE-sensitive gene loci and long telomeres disrupted upon telomere shortening. This looping mechanism was dubbed TPE-OLD for TPE over long distance [76].

To test whether changes in the higher-order organization of TPE-OLD genes correlate with epigenetics modifications, ChIP assay were used and revealed enrichment in H3K9me3 marks and, in the TRF2 shelterin component at the promoter of TPE-OLD genes indicating a possible cooperation between chromatin marks and telomeric factors in telomere-dependent gene silencing. TRF2 is specifically enriched at TPE-OLD sensitive genes, decreased upon telomere shortening and absent at the promoter of genes located within the loops. This suggests a gene-specific effect more than a global spreading of telomeric chromatin and heterochromatin marks.

Recently, a separate group explored telomere length influence through ChIP-Seq assays [101]. While confirming previous observations on TPE-OLD genes within 10 Mb from telomeres, they integrated genes further apart in their analysis and made striking observations. In short, the authors showed that TRF2 occupancy was telomere length dependent. TRF2 binds preferentially close to telomeres in cell with long telomeres and is relocated away from telomeres upon shortening (e.g., at ITS as far as 60 Mb from telomere). Gene expression and other epigenetic marks were modulated accordingly (e.g., decreased expression and H3K4me marks, increased in H3K27me3). Taken together these studies shed lights on additional mechanisms related to telomere length, emphasizing the potential role and genome-wide influence of telomeres [72]. More studies are required, using less artificial cellular models (e.g., cancer cells, telomere elongation using drugs and constructs) and more physiological validation (e.g., biopsies) to conclude and decipher the impact of this process.

TPE-OLD has been involved in other mechanisms not linked to telomere attrition but as an active mechanism involved in the regulation of genes during physiological responses [102], cancer cell differentiation [42,102] and even suggested as part of growth regulation during development (Romanov et al., in press). TPE-OLD is thus a mechanism depending on telomere length but also, more directly, on chromosome folding which can be envisioned as a regulatory mechanism that modifies the transcriptome in a telomere length-dependent manner. During cell lifetime, gene expression can be modulated to adapt to the environment before telomeres become critically short.

### 4.4. Telomeric Repeat-Containing RNA

For years, the longstanding belief was that telomeres are transcriptionally silent. Recent evidences show that telomeres are transcribed by RNA polymerase II to give rise to a class of long non-coding RNA containing telomeric repeats called TERRA (Telomeric Repeat-containing RNA) [77,103]. These transcripts have been detected in a variety of organisms, from yeast to Human. TERRAs are transcribed from subtelomeres, in a centromere to telomere direction and consist of 5′ to 3′ chromosome end-specific subtelomeric sequences and telomeric repeats. Direct visualization of TERRA by RNA-FISH revealed clusters and discrete foci of TERRA transcripts with some of them colocalizing with telomeres [77,103]. Association between these non-coding RNAs and telomere-binding protein was also shown, among which shelterin components such as TRF1 and TRF2 [104,105,106]. Mechanisms underlying TERRA function in telomere maintenance and regulation have been widely studied in budding yeast [107]. However, studies of potential TERRA functions in human cells show conflicting results between research groups, largely due to the challenges inherent to telomeres (e.g., length heterogeneity) and TERRA identification (e.g., no sequences consensus) [108,109]. For instance, whether TERRA molecules originate only from telomeres or also from internal telomeric sequences (ITS) is still debated. In addition, the question remains as whether TERRA molecules are produced from all chromosome ends and whether TERRA remain bound to the telomere from which they are transcribed or redistributed within the nucleus.

Telomeric RNAs are also implicated in R-loops formation which are RNA:DNA hybrids in which TERRA act either in *cis* or *trans* and suggested to interact with telomerase hence leading to different activities when occurring at telomeres or short/damaged telomeres [108,110,111]. RNA emanating from telomeres are also embroiled in pathologies like in the immunodeficiency, centromeric region instability, facial anomalies syndrome (ICF) [112].

## 5. Telomeropathies

Telomeropathies usually refer to diseases caused by defects in factors involved in the telomere maintenance machinery such as Dyskeratosis congenital, Revesz, Coat Plus or Hoyeraal Hreidarsson syndromes [113,114,115]. Telomere dynamics is profoundly affected in these diseases due to haploinsufficiency of components of either the telomerase holoenzyme (DKC1, TERT, and TERC), the TCAB1 shuttling factor, SnRNP components (NHP2 and NOP10 required for telomerase activity), the TIN2 shelterin component or the regulator of telomerase length helicase, RTEL1 [116,117,118,119]. However, unrelated syndromes involving changes in subtelomeres or telomeres might also be considered as telomeropathies [115]. Indeed, despite its protective role, telomere and chromatin environments are also involved, directly or indirectly, in different human genetic diseases.

Chromosomal translocation of region encompassing whole gene loci and thus implicating modification of the chromatin environment in gene expression modulation were observed in numerous genetic diseases. In the context of these so called “position effect mutations” modification in gene expression can be the consequence of two mechanisms. First, the chromosomal abnormality can separate the coding sequence from its regulating enhancers or silencers. Second, in case of translocations, the gene can be delocalized to a region harboring a different chromatin state. In both cases, a gene normally repressed can be activated and vice versa [120]. In the majority of these diseases, position effect mutations can lead to slightly different phenotypes from that caused by coding regions mutations. Indeed, position effect mutations can affect gene expression only in a subset of tissues or produce a milder phenotype but in most cases, the underlying mechanism is not established.

Non-physiological modification of telomere and chromosome ends integrity has also been associated to a number of diseases. Some patients diagnosed with various malformations, mental impairment, and growth retardation display formation of a ring chromosome. Such structures can occur on any autosomes and are formed by deletion on the short and long arms of the chromosomes followed by fusion at the breakage point. Consequences of ring chromosomes depend on different parameters: the chromosome implicated, the number of genes deleted, and the stability of the ring chromosomes. Most ring chromosomes are associated with loss of genetic material but in some rare cases, only telomeric regions are lost resulting in fusion of residual telomeres from both ends, forming a single telomere in the ring. This mechanism has been observed in ring 14, 17, and 20 syndromes [121,122,123]. Patients harboring ring chromosomes with telomere-telomere fusion showed downregulation of genes of the implicated chromosomes suggesting changes in the chromatin architecture. Indeed, telomere-telomere fusion can lead to the formation of a unique long telomere, enhancing telomere position effect into the ring chromosomes linking ring syndrome clinical phenotype to TPE-sensitive genes.

One of the best characterized human genetic diseases implicating TPE is Facio-Scapulo-Humeral Dystrophy (FSHD) [90]. A particularity of this disease is the late onset, with appearance of the first symptoms around the second life decade. The locus associated to FSHD is in the subtelomeric region of the long arm of chromosome 4 (4q35 locus) raising the possibility of TPE implication in FSHD pathophysiology. The leading candidate gene for FSHD pathogenesis is the DUX4 homeobox protein encoded by the last D4Z4 macrosatellite at the 4q35 locus. In FSHD patients presenting a diminished number of D4Z4 repeats (1–10 in FSHD1 patients versus 11 to 150 in non-affected individuals), relaxation of the D4Z4 array chromatin structure is associated with *DUX4* activation.

In this disease, the impact of telomere shortening was investigated in immortalized myoblasts expressing a floxable *hTERT* transgene [76,90]. In myoblasts harboring different telomeres size, *DUX4* expression was upregulated upon telomere shortening in FSHD cells, with a progressive effect as telomeres shorten, long before replicative senescence. TPE also extended to the upstream *FRG2* gene, located 70 kb more centromeric to *DUX4*, but was less prominent and not observed for *FRG1* located 90 kb upstream of *FRG2*. These observations uncovered FSHD as the first disease linked to classical TPE [90,124]. Using Hi-C experiments and 3C or 3D-FISH validation, long-range interactions between the *FRG1* and *SORBS2* locus were also identified in myoblasts from FSHD1 patients [125]. Interestingly, this long distance loop is lost upon telomere shortening in FSHD cells but not in control cells and correlates with *SORBS2* upregulation [125]. This suggests that telomere shortening on a D4Z4-contracted allele induces modification of the 4q35 folding with a more relaxed chromatin state. The *SORBS2* gene is silenced in normal cells by TPE-OLD mechanisms while this telomere-dependent mechanism is impaired in FSHD myoblasts with cooperation between the telomere and the repetitive macrosatellite array [125,126].

Another example of rare a disease involving telomeric changes is the immunodeficiency, centromeric region instability, and facial anomalies syndrome (ICF). ICF arises due to mutation in four different genes, *DNMT3B* (ICF1), *ZBTB24* (ICF2), *CDCA7* (ICF3), and *HELLS* (ICF4) [127,128,129,130]. Interestingly, while hypomethylation of subtelomeric regions, increased TERRA levels, telomere shortening and rapid senescence have been clearly evidenced in ICF1 [131,132], these hallmarks remained unchanged in ICF2-4 [133]. This again highlights the cross talks between the chromatin structure of telomeres and subtelomeres and further indicates that subtelomeric heterochromatin is regulated by specialized factors, the majority of which remain to be identified.

Furthermore, in agreement for a role of telomere subnuclear positioning in telomere regulation, telomeric defects are also observed in premature aging syndrome such as Hutchinson Gilford progeria linked to mutation in the *LMNA* gene encoding A-type Lamins [134]. In this disease and corresponding animal models, telomeres are shortened and mislocalized [135,136,137]. This is accompanied with ROS accumulation, an increased senescence and DNA damage response highlighting the existence of functional links between the different hallmarks of aging [138].

Telomere length is thus at the corner of different biological mechanisms but also associated to a broad range of diseases, from rare genetic syndromes to common ones. As we previously described, telomere shortening occurs normally during aging. This information has been broadly disseminated in the general population with an increasing number of newspapers mentioning the links between aging and telomeres length. As a consequence, numerous private companies offer telomere length measurement with the promise to “help you stay younger longer”, not mentioning cures or food supplement protecting telomeres.

A study published in 2009 aimed at establishing correlations between telomere length, causes of death and years of healthy life (YHL) by applying different statistical analysis on large population cohorts to prove correlation between these three factors [139]. The only positive and significant correlation was made between telomere length and YHL, suggesting that people with longer telomere could appreciate longer healthy aging than others. In agreement, telomere shortening has been associated with a number of age-related diseases such as cardiovascular diseases, dyslipidemia [140], hypertension [141], atherosclerosis [142], stroke [143], cancers [144], or diabetes [145]. In the same line, a number of studies showed the benefit of a healthy and sportive life in order to support the interest of telomere measurement and the protective effect of long telomeres in healthy living however the causative relationships remain to be deciphered in depth.

If one can criticize the statistical tests power, we now know the extreme variability of telomere length among individuals. Such studies validated telomere length as a good biomarker for healthy aging but interpretation must be taken with caution given the number of additional factors to be considered.

## 6. Conclusions

Topics on aging never get old and telomere biology is no exception. Telomere epidemiology studies already revealed that we are not “telomerically” equal and discoveries on telomere length regulation, heritability, dynamics and association with aging, raised interest on the topic to a general audience, from journalists to insurance companies. This peculiar part of our DNA has been at the center of interest for the past 80 years. In this context, it falls under the responsibility of the scientific community to convey a balanced message regarding the implication of telomere homeostasis in diseases and wellness, considering that numerous additional factors must be considered. Indeed, in human, the clear-cut influence of telomeres, beyond the well-described effect of critical shortening, remains unclear and record on telomere shortening rate at specific telomere-ends are missing, leaving the former observation to potential biased conclusions. Moreover, the recent identification of long non-coding RNA (TERRAs) produced by telomeres and the long-distance impacts of telomeres on gene regulation represents a new challenge. Thanks to the rise of high-resolution methods (TeSLA, STELA, HiC, molecular combing) and accessibility to affordable deep sequencing platforms (NGS, nanopore technology, PacBio), future research will help deciphering subtelomere complexity, likely uncover novel pathways linked to telomere biology and opens new research opportunities to understand the genome wide impact of telomere in physiological and pathological situations or adaptation to environmental cues. In this context, if all agree that telomere matters, questions on their influence, weakness and strength linger.

## Figures and Tables

**Figure 1 cells-08-00030-f001:**
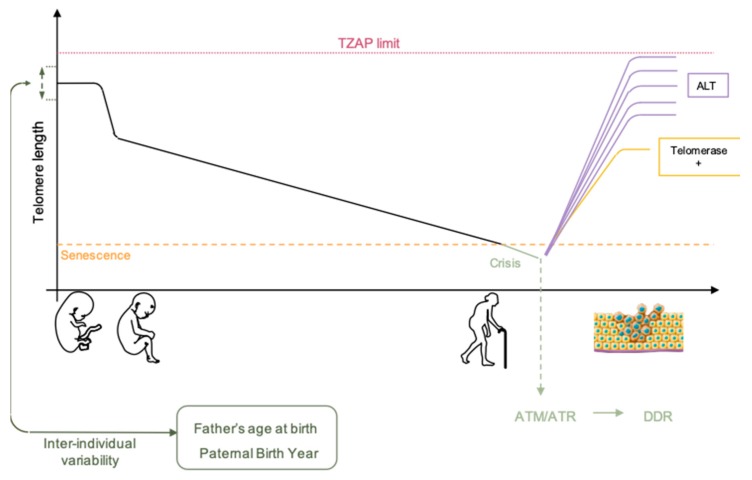
Telomere length decreases with age. Initial telomere variability comes from the parental telomere length and is modulated both by the father’s age at birth and paternal birth year, a correction factor allowing comparison between different datasets [38]. During development, telomere length decreases abruptly and reaches a steady slow rate of erosion after birth. Telomere attrition is confronted to a critical size corresponding to the senescence stage of the cell. Beyond senescence, cells undergo crisis with continuous telomere length decrease leading to activation of ATM/ATR checkpoint and DNA Damage Response (DDR) pathway activation. Cells surviving to crisis enter in an oncogenic process where telomere length increases thanks to telomerase reactivation or Alternative Lengthening of Telomeres (ALT). In all biological process, a telomere upper length limit is defined by TZAP, avoiding formation of extra-long telomeres.

**Figure 2 cells-08-00030-f002:**
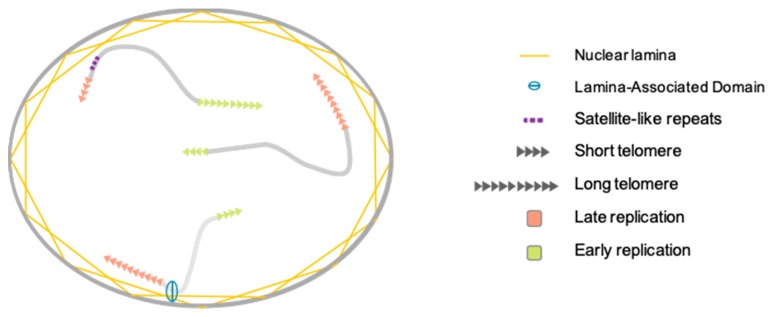
Telomere nuclear localization and replication timing. Similar to observations made on chromosomes, telomere replication timing is tightly linked to its nuclear localization. In brief, an internal localization correlates with early S-Phase replication and inversely, peripheral localization is associated with late S-Phase replication. Peripheral positioning of telomere is observed for subtelomeres enriched in beta satellite repeats. Through interactions with Lamin-associated partners, a number of telomeres lie in the vicinity of topological domains associated to the lamina (LADs). Importantly, even if an integrative 3D vision of telomere positioning is lacking, telomere replication occurs independently of telomere length.

**Figure 3 cells-08-00030-f003:**
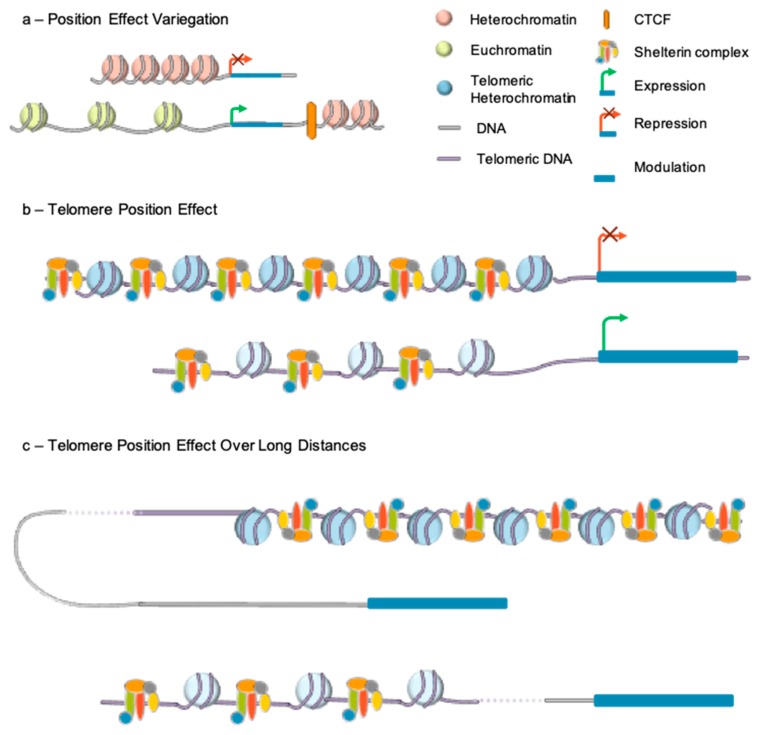
Gene transcription can be modulated by epigenetic features. (**a**) Position effect variegaton is caused by presence of heterochromatin regions in the vicinity of a gene. This repressive chromation environment is able to switch off gene expression by heterochromatin marks spreading. Spreading can be blocked by insulator elements bound by CTCF protecting the gene from heterochromatin. (**b**) Subtelomeric genes expression can be impeded by telomeric heterochromatin spreading through a mechanism called telomere position effect (TPE). Expression can be re-established as telomere shorten, either with age or in pathological conditions. (**c**) With chromosome folding, telomeres can loop over long distances, bringing together telomeric heterochromatin and gene located in more centromeric locus. Gene expression can be switched on or off, depending on a mechanism not yet fully understood termed TPE-OLD.

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
