# Peer review of "Bring It to an End: Does Telomeres Size Matter?"

_cells, 2019, doi:10.3390/cells8010030_

Round 1

Reviewer 1 Report

The review entitled "Bring it to an End" intelligently summarizes the recent findings regarding telomere length regulatory aspects as documented in the literature emphasizing TPE, TPE- OLD and TERRA. Although very comprehensive and critical, I have a few suggestions for improving it.

I would add a sentence describing telomerase already at the introduction, e.g. lines 118-121. In this way a brief explanation regarding why cancer cells usually possess short telomeres can be added, as telomerase upregulation is considered a late event during malignant transformation (At the end of line 47).

Line 61- please add "thus predicting the remaining cellular life span of the cell".

Figure 1- legend: the term "birth year should be explained here and in the relevant text. In addition, where beyond senescence is mentioned, the authors should explained that senescence is skipped upon dysfunction of the cellular checkpoints such as p53 or RB gene products.

Line 135: telomerase is constitutively expressed- albeit to a lesser extent in stem cells.  

After line 143 please add that these variance may reflect the global differences between the biology of telomeres and telomerase between humans and mice..

At the beginning of section 3.1, after ref. 59, 60, please explain what organisms were studied in this respect.

At the end of the "telomere replication" section, it is important to add a sentence explaining the significance of elucidating the time of telomere replication and the telomere nuclear localization.

Line 347- if the authors refer to mammals with regards to the DNA looping it is recommended to mention it there.

At the end of the TPE section (4.2) please give some examples regarding genes that were shown to be affected by it, at least in yeast, just to attribute some teleological "biological advantage" to this phenomenon.

After line 408- please elaborate shortly on the TERRA functions albeit they are conflictingly reported…

Section 5- please provide more examples on telomeropathies apart from FSHD

Minor small things:

Line 361: please add "possible" cooperation between chromatin marks…..

Line 83: has been associated with and not to.

In vitro/ in vivo: Italics

Author Response

With regard to reviewer 1

·      I would add a sentence describing telomerase already at the introduction, e.g. lines 118-121. In this way a brief explanation regarding why cancer cells usually possess short telomeres can be added, as telomerase upregulation is considered a late event during malignant transformation (At the end of line 47).

As suggested, a short sentence on the absence of telomerase in somatic cells was added in the introduction (lines 46-47): “From cancer research to ageing studies, investigations on the regulation of the very tip of chromosomes led to paradoxical observations: (i) telomerase, the enzyme responsible for telomere elongation is not produced in somatic cells, (ii) most cancer telomerase positive-cells display rather short telomeres as telomerase upregulation is considered a late event during malignant transformation; (iii) stem cells harbors long telomeres and express telomerase; (iv) short telomeres are associated with ageing and increased cancer risk, or the complete opposite; (v) telomere maintenance involves epigenetic mechanisms and chromatin changes.”.

·      Line 61- please add "thus predicting the remaining cellular life span of the cell".

The sentence has been modified as suggested.

·      Figure 1- legend: the term "birth year should be explained here and in the relevant text.

As suggested, the legend of the figure has been modified, together with the text (page 5): “As a result older fathers’ offspring inherit longer telomeres [56]. In a more recent study, using 3 of the largest available datasets, a group refined the calculations, and incorporated the paternal birth year as a corrective parameter allowing comparison of different datasets since Paternal birth year (PBY) was collinear with age and father's age at birth of the offspring in the three datasets used (e.g., considering a generational effect). Beyond reconciling data from cross-sectional and longitudinal studies, their outcome revealed that global telomere length shortens in the global population [57].”.

·      In addition, where beyond senescence is mentioned, the authors should explained that senescence is skipped upon dysfunction of the cellular checkpoints such as p53 or RB gene products.

This information has been added on page 2: “Telomere-induced senescence depends on the rate of telomere shortening, initial telomere length and length of the shortest telomeres [30] and can be skipped upon dysfucntion of the cellular checkpoints such as p53 or RB.”.

·      Line 135: telomerase is constitutively expressed- albeit to a lesser extent in stem cells. 

The sentence has been modified as suggested.

·      After line 143 please add that these variance may reflect the global differences between the biology of telomeres and telomerase between humans and mice.

This has been done as requested

·      At the beginning of section 3.1, after ref. 59, 60, please explain what organisms were studied in this respect.

As suggested the sentence has been modified: “For a long time, chromosome organization within the cell was thought to be random. In mammalian cells, the exact opposite is now well established [59,60] »

·      At the end of the "telomere replication" section, it is important to add a sentence explaining the significance of elucidating the time of telomere replication and the telomere nuclear localization.

The conclusion of this section has been expanded: “Thus, an integrative vision of telomeres within its 3D environment remains to be established to further understand the consequences of nuclear localization on higher order telomere organization and consequences on transcription or DNA repair of chromosome ends (Figure 2).”.

·      Line 347- if the authors refer to mammals with regards to the DNA looping it is recommended to mention it there.

The sentence has been modified as suggested: “If discontinuous TPE has been shown in yeast [95], its existence in mammals is fairly new. One hypothesis explaining TPE discontinuity in mammals might reside in DNA looping as involved in the enhancer-blocking activity or trans-acting changes occurring elsewhere in response to telomere shortening (Figure 3).”. Page 9.

·      At the end of the TPE section (4.2) please give some examples regarding genes that were shown to be affected by it, at least in yeast, just to attribute some teleological "biological advantage" to this phenomenon.

The end of the TPE paragraph has been modified as suggested: “These findings were consistent with yeast TPE mechanisms. In this specie, most subtelomeric genes are involved in metabolism and stress response and are silenced in optimal growth conditions (Stone, Pillus JCell biol 1996). These genes, such as PAU involved in cell wall (Ai, Bertram Mol cell 2002) and drug resistance or FLO involved in cellular adherence (Halme et al. Cell 2004) are activated upon stress, nutrient starvation of chemical treatment. Interestingly, as observed in budding yeat, ISG15 is involved in the innate immunity response pathway suggesting a conservative role of TPE [94], as a stress-response mechanism. “.

·      After line 408- please elaborate shortly on the TERRA functions albeit they are conflictingly reported…

We further developed this part by adding the following information on page 10: “For instance, whether TERRA molecules originate from telomeres or also from internal telomeric sequences (ITS) is still debated. In addition, the question remains as whether TERRA molecules are produced from all chromosome ends and whether TERRA remain bound to the telomere from which they are transcribed.”.

·      Section 5- please provide more examples on telomeropathies apart from FSHD.

This section has been extended. We now shortly list different telomeropathies and diseases in which telomeres might be implicated such as Ring chromosome syndrome, FSHD, Progeria and ICF syndrome.

Minor small things:

Line 361: please add "possible" cooperation between chromatin marks…..

Line 83: has been associated with and not to.

·      In vitro/ in vivo: Italics.

This has been changed throughout the text.

Reviewer 2 Report

Bring it to an End:

This is an excellent review on telomere biology by Laberthonniere and coauthors.

It focusses mainly on molecular aspects of telomeres, rather than clinical implications. The authors are very competent in this area of biology, hence the review is written at a very high standard. I only have some minor points to address:

1.    Some passages need to be edited by a native speaker; there some typographic mistakes (Figure 3: Position Effect/Variegation) and grammar issues, e.g. Figure 3: Gene expression can be switch on or off (switched), line 435 ‘This mechanisms’ (instead of these mechanism), and multiple others

2.    How can novel technologies help to look more precisely at telomere length and subtelomeric areas. Is deep sequencing suitable to quantify telomere length more accurately ? 

3.    What exactly is the molecular link between laminopathies and telomere length ?

4.    The authors suggest that although short (mean) leukocyte telomere length is associated with inflammatory age-related diseases, there is no causal relationship; specifically ‘what is the consequence of short telomeres from a molecular point of view ?’ 

5.    How exactly do telomeres shorten in non-mitotic tissues ?

Author Response

With regard to reviewer 2 

1.    Some passages need to be edited by a native speaker; there some typographic mistakes (Figure 3: Position Effect/Variegation) and grammar issues, e.g. Figure 3: Gene expression can be switch on or off (switched), line 435 ‘This mechanisms’ (instead of these mechanism), and multiple others

We apologize for these errors and the text has been edited carefully

2.    How can novel technologies help to look more precisely at telomere length and subtelomeric areas. Is deep sequencing suitable to quantify telomere length more accurately ?

We were referring to different methodologies for telomere length analysis but also subtelomeric variability. The sentence has been modified to address Reviewer 2’ comments: “Thanks to the rise of high-resolution methods (TeSLA, STELA, HiC, molecular combing) and accessibility to affordable deep sequencing platforms (NGS, nanopore technology, PacBio), future research will help deciphering subtelomere complexity, likely uncover novel pathways linked to telomere biology and opens new research opportunities to understand the genome wide impact of telomere in physiological and pathological situations or adaptation to environmental cues.

3.    What exactly is the molecular link between laminopathies and telomere length ?

A short paragraph describing telomere alterations in laminopathies has been added in the “Telomeropathies section” (section 5).

Furthermore, in agreement for a role of telomere subnuclear positioning in telomere regulation, telomeric defects are also observed in premature ageing syndrome such as Hutchinson Gilford progeria linked to mutation in the LMNA gene ecoding A-type Lamins (De Sandre,; Erickson). In this diseased and corresponding animal models, telomeres are shortened and mislocalized (Taimen et al 2009; Gonzalez-Suarez et al 2009; redwood et al 2011; Redwood et al 2011) and is accompanied with ROS accumulation, an increased senescence and DNA damage response highlighting the existence of functional links between the different hallmarks of aging.”

4.    The authors suggest that although short (mean) leukocyte telomere length is associated with inflammatory age-related diseases, there is no causal relationship; specifically ‘what is the consequence of short telomeres from a molecular point of view ?’

This comment relates to the end of paragraph 6 that we have slightly modified by changing the following sentence: “In the same line, a number of studies showed the benefit of a healthy and sportive life in order to support the interest of telomere measurement and the protective effect of long telomeres in healthy living however the causative relationships remain to be deciphered in depth. “

5.    How exactly do telomeres shorten in non-mitotic tissues ?

This is indeed an interesting question which has not been resolved so far. As suggested, this point is now discussed at the end of section 2: “Furthermore, in agreement for a role of telomere subnuclear positioning in telomere regulation, telomeric defects are also observed in premature ageing syndrome such as Hutchinson Gilford progeria linked to mutation in the LMNA gene ecoding A-type Lamins (De Sandre,; Erickson). In this diseased and corresponding animal models, telomeres are shortened and mislocalized (Taimen et al 2009; Gonzalez-Suarez et al 2009; redwood et al 2011; Redwood et al 2011) and is accompanied with ROS accumulation, an increased senescence and DNA damage response highlighting the existence of functional links between the different hallmarks of aging.”.